# Barriers and Facilitating Factors of Adherence to Antidepressant Treatments: An Exploratory Qualitative Study with Patients and Psychiatrists

**DOI:** 10.3390/ijerph192416788

**Published:** 2022-12-14

**Authors:** Beatriz González de León, Analía Abt-Sacks, Francisco Javier Acosta Artiles, Tasmania del Pino-Sedeño, Vanesa Ramos-García, Cristobalina Rodríguez Álvarez, Daniel Bejarano-Quisoboni, María M. Trujillo-Martín

**Affiliations:** 1Multiprofessional Teaching Unit of Family and Community Care La Laguna-Tenerife Norte, Management of Primary Care of Tenerife, 38320 Santa Cruz de Tenerife, Spain; 2Canary Islands Health Research Institute Foundation, 38320 Santa Cruz de Tenerife, Spain; 3Evaluation Unit (SESCS), Canary Islands Health Service (SCS), 38109 Santa Cruz de Tenerife, Spain; 4Network for Research on Chronicity, Primary Care, and Health Promotion (RICAPPS), 38109 Santa Cruz de Tenerife, Spain; 5Department of Mental Health, General Management of Healthcare Programs, Canary Islands Health Service, 35071 Las Palmas de Gran Canaria, Spain; 6Research Network on Health Services for Chronic Conditions (REDISSEC), Carlos III Health Institute, 28029 Madrid, Spain; 7Department of Psychiatry, Insular University Hospital of Gran Canaria, 35016 Las Palmas de Gran Canaria, Spain; 8Department of Preventive Medicine and Public Health, University of La Laguna, 38071 Santa Cruz de Tenerife, Spain; 9Center for Public Health Research (CSISP-FISABIO), 46020 Valencia, Spain; 10Institute of Biomedical Technologies, University of La Laguna, 38200 Santa Cruz de Tenerife, Spain

**Keywords:** medication adherence, depressive disorders, antidepressants, qualitative research

## Abstract

This study examines the experiences and expectations of patients with depressive disorders regarding the disease and different antidepressants, as well as examining the barriers and facilitating factors that could affect their adherence to medications. An exploratory qualitative study was carried out. The study involved two focus groups made up of patients and caregivers and six semi-structured interviews with psychiatrists. In both cases, the participants were selected by intentional theoretical sampling, seeking maximum significance variation of social types. Prejudice about the side effects of medication was relevant. The importance of patients being well informed about the disease/treatments was noteworthy. The stigmatization of antidepressants by patients was identified as a barrier to medication adherence. The involvement of family members and the motivation of patients to be actively involved in the process to recover from the disease were identified as facilitating factors. The work carried out suggests the need for patients to have rigorous information about the disease/treatment to reduce the possible prejudices generated by beliefs. Maintaining greater contact and monitoring of patients/caregivers to help therapeutic adherence in patients with depressive disorders was also identified as being of great importance.

## 1. Introduction

Depressive disorders affect more than 280 million people worldwide and are considered one of the main causes of disability [1] and contribute to the global burden of disease [2]. Therefore, these disorders produce a significant health cost [3].

Antidepressant drugs are still a widely used option among the treatments available for people with depressive disorders [4,5]. The proper prescription of these medications is key in modern mental health care not only to achieve response and remission but also to prevent possible relapse [6]. However, the effectiveness of drug therapy depends not only on the theoretical efficacy, diagnostic adequacy, and suitability of the active compounds used but also on the patient’s adherence to the prescribed regimen [7,8]. Although the rate of early adherence to antidepressant medication is estimated to be between 74% and 82%, approximately 50% of patients discontinue it prematurely [9,10]. Even though 49–84% of patients perceive the need for antidepressant treatment [11], a third of the patients stop treatment at three months after feeling better, with the abandonment rate increasing to 55% at six months [12]. Drug side effects contribute to this therapeutic discontinuation [11].

Non-adherence to antidepressants has a high cost for the individual and society since it is associated with an increased risk of progression in severity, relapse, and recurrence and higher rates of emergency visits and hospitalizations, accounting for up to 39% of the total care costs per patient [13]. It is for these reasons that the World Health Organization declared depressive disorders as one of the nine chronic diseases on which attention should be to focused to improve adherence [14].

Adherence to antidepressants seems to be a multifactorial phenomenon in which factors related to the patient, the disease, the medication, the physician and health care, the family, and friends, as well as media and society interact [15].

Innovation in interventions aimed at improving adherence need to be supported by a better knowledge and understanding of the complex nature of adherence and their different interactions with the patients’ beliefs, knowledge, and expectations regarding depression and its treatment [16], levels of self-efficacy or psychological reactance [17], and health locus of control [18]. The complexity of this phenomenon requires (as in other scenarios in which a modification and consolidation of behaviors is sought) complex or multicomponent interventions supported by the use of information and communication technologies [14]. Multicomponent interventions concerning patients and/or physicians addressing structural aspects of care are more effective in improving medication adherence than simple single-component interventions [19].

The present study was carried out aiming to examine the experiences and expectations of patients toward antidepressant drugs and to identify the possible factors (barriers and facilitators) that could condition the adoption by psychiatrists of interventions focused on improving adherence to medication. To the best of the authors’ knowledge, few qualitative studies with this aim have been conducted to date [20,21]. This study was conducted as a part of a wider research project whose main objective is to assess the effectiveness and cost-effectiveness of a multicomponent strategy to improve adherence to pharmacological treatment in patients suffering from depression [22].

## 2. Materials and Methods

### 2.1. Design

An exploratory qualitative study was carried out by means of focus groups (FG) conducted with patients and caregivers alongside semi-structured interviews with psychiatrists. The FG was the chosen technique to collect relevant information from patients through a dynamic group interaction, allowing discussion and shared reflection about ideas, opinions, perceptions, attitudes, and beliefs based on participants experiences in a defined area of interest (their depressive disorder, the relationship with professional care, and the medication treatment), while the individual semi-structured interview was chosen as the appropriate technique to perform an in-depth investigation into the perceptions and experiences of the psychiatrists regarding the patients who presented difficulties in medication adherence. Interviews and FG were facilitated by a professional with qualitative research experience from a health anthropology background.

### 2.2. Participants Selection

Purposive sampling was used to ensure the sample was broadly diverse in regard to age, gender distribution, time since diagnosis, and pharmacological treatment [23]. Participation was voluntary and written informed consent was obtained in all cases.

a.Semi-structured interviews

Participants were psychiatrists performing their professional practice in a Community Mental Health Unit (CMHU) in the Canary Islands (Spain). The final number of informants was defined sequentially and cumulatively, depending on the quality of the feedback and the saturation of the information.

b.Focus groups

Potential participants were identified by the abovementioned psychiatrists. Inclusion criteria were patients 18–65 years old (and/or their informal caregivers) with a diagnosis of depressive disorder (ICD-10: F32-F33) or dysthymia (ICD-10: F34.1), taking antidepressants, and with sufficient cognitive abilities to participate. Those patients with severe psychiatric illness, in the remission phase, who had not attended a medical visit in the previous six months, and/or with some physical disability that limited participation in group activities were excluded. 

Recruitment took place between July and September 2020. Participants were invited to opt-in to the qualitative study by research staff via a telephone call. Subsequently, those who agreed to participate were sent the study information sheet and signed the informed consent form.

### 2.3. Data Collection

a.Focus groups

The FG discussions were conducted according to a semi-structured guide developed from discussions among the researchers and a review of the main preliminary studies. The guide included the following topics: (a) experience of depression, (b) information received in the care process, (c) doctor-patient relationship, (d) impact of pharmacological treatment, and (e) adherence to anti-depressive medication (see Appendix A). The FG with patients from Gran Canaria was carried out online. In this case, the same person acted as observer and moderator. In the case of the Tenerife group, this was conducted face to face and there was one person who acted as moderator and another person who acted as observer. The discussions were audio recorded and transcribed for later analysis. Transcripts were reviewed for accuracy, and names and other identifiers were removed.

b.Semi-structured interviews

A list of interview topics was sent to the psychiatrists before the interview so that they could prepare their answers. Topics were: (a) experience in managing depression, (b) difficulties in treating depression, and (c) barriers and facilitators regarding adherence to pharmacological treatment (see Appendix B).

The interviewer had no professional or private relationship with any of the participants and the interviewer took notes during the interviews. All interviews were audio recorded and transcribed for later analysis. Transcripts were reviewed for accuracy, and names and other identifiers were removed.

### 2.4. Analysis

Transcripts were translated from Spanish to English by a professional translator. As meanings of participants’ experiences is conveyed through language and is mediated by sociocultural context, a native English translator, who has lived in the Canary Islands for more than thirty years, reviewed the transcripts to check for differences in the equivalence of meanings between the two languages and made modifications when deemed necessary. Any interpretation doubts that arose were consulted with the research team.

A thematic content analysis of transcripts was performed. Manual coding was performed to code the main themes of the interviews and focus groups. The analysis took the form of the following three stages: the coding of the findings of primary material; the organization of these ‘free codes’ into related areas to construct ‘descriptive’ themes; and the development of ‘analytical’ themes [24]. Two researchers (BGL and AAS) independently coded the text. The thematic categories were agreed upon through discussion and a framework informed by their open coding, and the study objectives were collaboratively developed. This first framework was then adapted from a previous framework [14,25]. Due to the professional profiles of the researchers who performed the coding and analysis (clinical experience as a primary care [PC] physician and the medical anthropologist with experience in qualitative evaluation of health services), there were differing views about the causes of depressive disorders and their treatment. The process of reflexivity on the preconceptions that could be conditioning the analysis in both researchers was essential in making these preconceptions explicit and correcting possible biases, resulting in a dialogue that enriched and complemented the jointly performed analysis.

The results are presented anonymously according to ethical criteria and data protection of the participants.

## 3. Results

Two FGs were conducted with seven and four patients each. The first group took place in October 2020 in the island of Gran Canaria, and the second in December 2021 in the island of Tenerife. The semi-structured interviews were in June–August 2020. 

### 3.1. Characteristics of Participants

In the first FG, two of the participants were informal caregivers (women) of two patients, who also participated. In addition, two were men (50 and 58 years old), and five were women who had a mean age of 44 (SD 5.40, range 38–52). In the second FG, three participants were women and had a mean age of 58.33 (SD 9.87, range 47–65), and the only man was 64 years old. Regarding the characteristics of the patients, most of them (N = 8, 89.89%) had a diagnosis of major depressive disorder and one had dysthymic disorder (11.11%), with a mean duration since diagnosis of the disease of 5.47 (SD 6.65) years. Of the nine patients, eight had between one and six years of treatment and one had twenty-two years of treatment. They were being treated with a mean of two antidepressant drugs (range: 1–4), and most of them presented low medication adherence. Table 1 and Table 2 show the general and individual sociodemographic as well as the clinical characteristics of the patients and caregivers. 

Six psychiatrists participated in the semi-structured interviews, three of whom were women, and they had a mean age of 39.43 (SD 11.83, range 39–63) (see Table 3). Two of them exercised their profession in Tenerife and the other four in Gran Canaria.

### 3.2. Perception of Patients and Professionals Regarding the Disease and Its Treatment

#### 3.2.1. Patient Experience of Depression

Patients said they felt misunderstood or undervalued during their illness. They had the feeling that those around them did not perceive the disease with the level of severity that corresponds to it. 


*“I think depression must be differentiated from having a specific bad spell, (…) but hardly anyone understands depression.”*
(Patient 2)


*“It is a disease that is not diagnosed with a blood test, so people do not understand it.”*
(Patient 4)

The keys that could favor recovery are family support and information. They pointed out that experts should communicate what is happening to family members, so they understand it better.


*“(…) But, we really need someone to help family members understand what we’re going through.”*
(Patient 2)


*“For me, information is essential and the support of relatives, too.”*
(Patient 7)


*“Information to relatives is very important so that they understand what is happening.”*
(Patient 8)


*“Sometimes, when you get a very severe depression you don’t recognize yourself, nor do your relatives recognize you, because you seem to be another person. The information in this moment is important.”*
(Patient 6)

Another aspect that they consider necessary for recovery are professional interventions and medication. They consider that the support of professionals is essential for recovery.


*“I have had depression before and I got better on my own, but in this case, I needed medication. It was impossible to get better alone; therefore, medication was essential in this case.”*
(Patient 1)


*“I think that medical support is essential. It (depression) is something that we do not understand, we cannot find an explanation for it, and so you have to seek help from people who are trained in the matter, who know, to start getting out of it. Then, of course, you must also do your bit, that is, you cannot wait for the medication to take effect, but you must also put in a little on your part.”*
(Patient 7)


*“The help of a psychiatrist has been very good for me. The advice he gives you, the guidelines you can follow.”*
(Patient 8)

#### 3.2.2. Need of Information

Most patients report having received the information about treatment correctly from professionals. 


*“Yes, on the part of the psychiatrist, yes.”*
(Patient 3)


*“I received all the information. He told me what each thing was for, how I was going to take it, how I was going to start, and he explained everything to me… He explained everything to me, the entire medication process.”*
(Patient 8)

Although some participants recognize that they have not always received information correctly because the psychiatrist had not provided it or because the patient did not show any interest at all. On the professionals’ side, they highlight the questionable quality of the information sources used by patients. 


*“Yes, people search on the internet. Another thing is that they search on the correct website.”*
(Psychiatrist 5)


*“It’s a bad sign when a patient comes with what the neighbor told him/her. That’s a bad sign, it’s worse than googling.”*
(Psychiatrist 5)


*“My experience is that it depends on the information. Many times, it is not entirely correct or the websites where patients find information are not entirely reliable. Then, they come with wrong expectations.”*
(Resident doctor 1)

Firstly, professionals do not think there is a lack of correct information available to patients.


*“The information is accessible (…). On professional websites, Spanish Society of Psychiatry and Ministry of Health, there are specific guides on depression. In other words, access is not the problem.”*
(Psychiatrist 5)

Secondly, some psychiatrists consider that their work is easier when patients are well informed:


*“I prefer informed patients, well-informed patients, because I think that teamwork is necessary. At least, we are two experts.”*
(Psychiatrist 5)


*“I prefer they come informed, and they consult with me about their doubts, because they can sometimes come misinformed…”*
(Psychiatrist 1)

However, other professionals report not feeling comfortable managing the treatment with the patient who is very informed:


*“I don’t prefer the savvy patient, who knows everything, who has read up on it… I like to explain a little bit to them, but today, when you mention serotonin, they say ‘yes, yes, serotonin sounds familiar to me’.”*
(Psychiatrist 2)


*“I admit that it’s sometimes easier for me if they come and don’t object… I’m a bit ambivalent about that.”*
(Resident doctor 1)


*“New patients (patients who have not received information) who come for the first time to receive a treatment generally do not cause problems; I explain the treatment and that’s it.”*
(Psychiatrist 5)

#### 3.2.3. Treatment of Depression

##### 3.2.3.1. Depression Management and Treatment Decision

Health professionals agree that there is a wide spectrum of levels of depression, with mild or moderate depression requiring the least effort. Some aspects that are considered to involve greater complexity are suicidal ideas, anhedonia, and apathy. 


*“Severe depression depletes resources, but when it is a mild or moderate depression, it requires a normal effort.”*
(Psychiatrist 4)

By contrast, depression associated with other problems involves more complex management. It should be mentioned that the greatest difficulty in management does not lie in the pharmacological treatment, but rather in the patient’s characteristics.


*“Real depressions are those that require less work. On the other hand, depressions with personality disorders, with family and social problems are the ones that require the most work.”*
(Psychiatrist 4)


*“An endogenous depression hits you and you want to cure yourself, like any other disease. People who have an underlying neurosis do not always want to be cured, and the most difficult thing is being able to cure them when they don’t want to be cured; it’s quite complicated.”*
(Psychiatrist 5)

This is why psychiatrists think that they should prescribe drugs associated with the individual needs of patients.


*“You have to be very meticulous, identifying what the characteristics of the patient are in all senses, that is, in the personal sphere, in the field of health, what are the other pathologies that they have.”*
(Psychiatrist 3)

Additionally, professionals consider that the patient’s personal medical history, such as the type of treatment they have previously had, can influence the difficulty of managing this disease.


*“The problem is that many of the patients are resistant (to treatment). (…) So, first, you have to investigate more, especially the psychopharmacological approach that has been used.”*
(Psychiatrist 1)

At the same time, other psychiatrists consider that the drug regimen is protocolized, and the difficulty is found in psychotherapy. Psychotherapy is perceived as necessary, although affected by the lack of time and professionals.


*“Pharmacological treatment has some protocols, that is, this part is not very complex”*
(Psychiatrist 2)


*“Those depressions with social, anxiety, family factors (…) are the ones that involve more work. In my opinion, due to the lack of psychological support.”*
(Psychiatrist 2)


*“Psychotherapeutic work is harder, that is, with the person, trying to understand their situation… These activities are more typical of psychology, but the conditions are not met, especially in the hospital setting.”*
(Psychiatrist 2)


*“Psychologists, due to their scarcity, cannot apply the therapies as they should and that ends up with overloading.”*
(Psychiatrist 2)


*“The system is not oriented so that the patient can receive regulated psychotherapy (because of a lack of psychologists).”*
(Psychiatrist 3)

However, the figure of the psychologist has not always been as well received by patients as that of the psychiatrist:


*“So, honestly, I didn’t like them (psychotherapists).”*
(Patient 8)


*“Every time I go to a psychologist, it’s like going to a slaughterhouse.”*
(Patient 3)

Psychiatrists said that on many occasions they have to do psychologists’ work, although they are not always trained well enough to do so:


*“In the end, the psychiatrist ends up trying to do psychotherapy, without being the right professional, because the right professional should be a psychologist to do therapy properly.”*
(Resident doctor 1)

Another psychiatrist considers that they do not have enough time to do psychotherapy:


*“I can’t do it with everyone (psychotherapy), I don’t have time.”*
(Psychiatrist 5)


*“I try, but I recognize that I cannot provide regulated psychotherapy (due to lack of time)”*
(Psychiatrist 4)

They consider that some tools are useful in clinical practice, such as motivational interviewing but that they need training to do it. However, they consider that sometimes the severity of the symptoms does not allow the tool to be used.


*“It (motivational interview) is useful, but (…) you have to be trained. It is true that I have been half trained in motivational interviews. The truth is that I need the second half.”*
(Psychiatrist 3)


*“If we are treating a very, very severe case of major depression, (…) we have to prioritize the drug treatment before using another type of strategy.”*
(Psychiatrist 1)

At least, on many occasions, professionals say that the lack of time is the main factor that prevents the tools from being correctly used on a daily basis.


*“If there is not enough time to attend to a patient, there is no motivational interview or anything, because you cannot dedicate yourself to them calmly.”*
(Psychiatrist 3)

When prescribing treatment, professionals say that it is important to consider patients’ preferences when choosing treatment. 


*“There are patients who ask you this (pseudoscience). I think that the patient’s decision must be respected as long as they are well informed.”*
(Psychiatrist 3)

However, it is not always considered a first-order necessity by the professional.


*“I would take into account the functionality of the patient, and by functionality, I mean work or daily living activities (…), and then I would also take into account the patient’s preferences.”*
(Psychiatrist 3)


*“This has to be done, I can’t tell you if it is done regularly, but it has to be done.”*
(Psychiatrist 3)

In addition, they say they have little time to inform patients about their treatment:


*“There is a lack of time for professionals to explain not only what is happening but also what the therapeutic possibilities are.”*
(Psychiatrist 3)


*“Well, it is true that I do not always do it perfectly. Sometimes the time is pressing and well…”*
(Psychiatrist 4)

Professionals feel that shared decision-making (SDM) is necessary because it considers the preferences of the patients and is an important part of their work.


*“I really like to involve the patient in treatment options.”*
(Psychiatrist 2)


*“Involving the patient is essential in the improvement process.”*
(Psychiatrist 4)


*“I think the decision is made by the patient, it is clear, no matter how many pills you prescribe.”*
(Resident doctor 1)


*“Many times, they leave the decision to us (professionals), even if you tell them, they tell yo, whatever you think is best, I don’t understand.”*
(Resident doctor 1)


*“There are many patients who are not very interested in that (SDM), but what they simply want is for the doctor to prescribe something and they don’t even get into discussing anything.”*
(Psychiatrist 1)

However, they also think SDM is not always applied. Therefore, it is not a very widespread process. Additionally, although there are detractors, professionals consider, once again, that lack of time is the main conditioning factor.


*“Few psychiatrists say, ‘we would have this and that treatment and you choose’.”*
(Resident doctor 1)


*“It is true that there are colleagues who feel more comfortable with paternalism: ‘you must take this because I say so’.”*
(Psychiatrist 3)


*“The time, the lack of time. If you have little time, you cannot apply shared decision-making because it requires time.”*
(Psychiatrist 3)

Some patients feel their doctor consults them and others do not. Some participants defended the paternalism of the professional, alluding to the fact that they did not have sufficient knowledge of pharmacology.


*“Yes, the psychiatrist and I were talking about it, and she listened to me. Then, we made the decision together.”*
(Patient 3)


*“In my case, he explained the pros and cons to me, but since it’s an issue I don’t understand, I take the medication, and then, if I feel bad, I ask him to change it.”*
(Patient 3)

Finally, professionals say that the role of PC in the first diagnosis and prescribed treatment is a very important factor in the pharmacological treatment decision. However, there is a disparity of criteria among the professionals. Although some of them justify this limitation because of the precarious circumstances of PC, other psychiatrists do not consider these PC circumstances and blame PC professionals for the work overload of psychiatrists.


*“I think the main problem is when the diagnosis is not correct. In other words, if you are scrupulous about the psychopathology of depression, you will apply a treatment that will be effective, but on many occasions, there are patients who are diagnosed with depression when they are not depressed. Generally, the first diagnosis is made by a PC physician (…) the diagnosis is not always the right one.”*
(Psychiatrist 3)


*“They (family doctors) have to cover many things. So, many times there is an underdiagnosis.”*
(Psychiatrist 3)


*“PC doctors have five or seven minutes per patient. Many times, they prescribe a standard medication. However, when the patient is treated by a psychiatrist, then we have more time to get to know the patient, and the treatment is more tailored.”*
(Psychiatrist 3)


*“I think that a PC physician prescribing an antidepressant treatment for 6–12 months before referring them to a psychiatrist is too much.”*
(Psychiatrist 4)


*“There are depressions that are very easy to treat and that is also another reason for overloading psychiatrists with work. (…) There are a lot of patients who are not even minimally treated by their family doctors before referral to a psychiatrist.”*
(Psychiatrist 2)

##### 3.2.3.2. The Perception of the Impact of Medication and the Patient’s Treatment Experience

No participant wants to take pharmacological treatment. Rather, they wish to stop it as soon as possible, mainly because of the impact it has had on them. 


*“I think that no one likes to take medicine, no one. You take it because you have no other solution.”*
(Patient 4)


*“I think that if you read the prospectus, you won’t take the medication, but, of course, if you don’t take this medication, there is often no other option.”*
(Patient 3)

With the same unanimity, the patients perceived that this was sometimes non-negotiable and necessary for recovery. 


*“In my case, medication was positive to stop having dark and obsessive thoughts and I imagine that without medication I would not have been able to do it.”*
(Patient 3)

Although on some occasions they have even thought it was dangerous.


*“I think medication has some danger, at least in my case it has…, and what I mean is that I have had the intention and desire to take all the medication at the same time, the entire bottle.”*
(Patient 4)

Another noteworthy point is that professionals think patients usually communicate the problems they encounter with their pharmacological treatment. This has not been corroborated by the patients.

### 3.3. Modifying Factors of Pharmacological Adherence

#### 3.3.1. Factors Related to the Health System

##### 3.3.1.1. Doctor–Patient Relationship

Patients and psychiatrists agree that one of the main facilitators of adherence is the doctor–patient relationship, especially the trust in the doctor.

##### 3.3.1.2. Coordination with PC

By contrast, professionals have identified coordination with PC as a determining factor in the management of patients with depression. 

##### 3.3.1.3. Accessibility to and Availability of Professionals

Another important aspect is accessibility to talk about doubts with the professional. This aspect is identified by both patients and professionals. However, among all the possible factors related to the system and the organization, time is the most prominent. This point is identified by psychiatrists who reiterate this lack of time many times during the interviews. This lack of time makes it difficult for professionals to work on adherence. As well as the lack of psychologists conditioning the psychotherapy offered to patients.

##### 3.3.1.4. Information Available to the Patient

As regards this issue, professionals highlight the importance of giving adequate information to the patients. In addition, the need to adequately inform the patient about the diagnosis, prognosis, and treatment options has been identified. During the interviews, the psychiatrists implicitly stress that the type of approach used for these patients is pharmacological, with psychotherapy being thought of as the territory of the psychologist.

##### 3.3.1.5. Shared Decision-Making and Other Methods

In the field of therapeutic decisions, even though not everyone puts it into practice, professionals and patients consider SDM to be important in medication adherence. In addition to the implementation of tools, such as SDM, the use of tools that motivate the patient have been mentioned (such as motivational interview). Multicomponent methodologies and strategies are also identified by professionals as facilitators of adherence. 

##### 3.3.1.6. Factor Related to Patients and Socio-Family Context

Patients have identified forgetfulness as a barrier to adherence. They say that having a system that reminds them to take medication can be useful. Additionally, none of the patients say they want to take drugs, but those who take them have the hope of returning to how they were before.

By contrast, professionals point out that patients who are more aware of the disease are more adherent. Additionally, psychiatrists identify the disease severity as another factor, which may affect adherence, as much as being confident in the medication’s effectiveness. However, the perception of disease control may involve discontinuation of the medication. Feeling better when they have just started treatment and the difficulty in keeping a schedule is identified as a barrier. The time between consultations with the specialist is also mentioned by patients as a factor for discontinuing the medication.

Another factor identified as a facilitator by psychiatrists and patients is social/family support. In addition, the professionals interviewed said that social opinion concerning the treatment may influence the patient’s opinion. Moreover, psychiatrists consider that if the family is well informed, they will be able to better address their disease and its management.

#### 3.3.2. Factors Related to Depression and the Pharmacological Treatment Itself

Patients identify the delay in the onset of treatment effect as a limitation for adherence, and professionals identified prejudices about side effects of the treatment as the main negative factor concerning adherence. This barrier has also been identified in some patients.

Nevertheless, in some cases, side effects really do occur in patients. However, some patients reported not worrying about the side effects since they are more afraid of the disease itself. 

Table 4 presents a more detailed description of the factors that modify pharmacological adherence with the quotes of the participants.

## 4. Discussion

The results of the present exploratory study illustrate the experiences and expectations of patients with depressive disorders and identify barriers and facilitators for their adherence to pharmacological treatments and those that could condition the adoption by psychiatrists of interventions focused on improving adherence to medication in these patients. The data analysis was performed by two researchers with different professional profiles. One of them works as a clinician and the other as an evaluative researcher. Despite the differences, the reflexivity process was able to minimize the biases of each one and has enriched the analysis and has been key to developing an intervention strategy from a more complex approach. Traditionally, the model of causality assumed in the professional training of clinician and anthropologist is different. While it is more frequent for the former to have training with a traditional linear causality approach between cause and effect, the latter is more familiar with a complexity approach, where causality emerges from multiple influences that interact to arrive at a particular result [26]. These views and these assumptions have emerged during the discussion of the results among the researchers and an effort has been made to maintain a complex view of causality in the conclusions.

Taking antidepressants was identified by patients as a factor influencing clinical improvement. This finding coincides with the results of previous studies [27]. Patients do not want to take drugs chronically, but at the same time, they consider that medication is necessary for recovery. These results can be found in other studies in which chronic pharmacological treatments are perceived as a burden and a loss of autonomy [25]. Moreover, it should be noted that patients also reject psychotherapy. This refusal of any type of treatment (pharmacological or non-pharmacological) could be related to the idea of identifying themselves as sick people, so the patient may interpret this situation with a reduction in the capacities and possibilities [28]. However, on other occasions, patients consider treatment as an opportunity to recover their health status and improve their quality of life.

Beliefs about side effects were identified as a barrier, both by patients and professionals. This aspect appeared repeatedly in the discourse, mainly in the case of the professionals, which highlights its importance. In this regard, professionals consider that patients have erroneous and preconceived ideas about the side effects of antidepressant drugs. By contrast, patients recognized that they are concerned about these side effects, and these concerns affect their adherence. In depression, the side effects of medication have been associated with non-adherence [29], especially those related to sleep, and the digestive and nervous systems. However, although widely known side effects can certainly occur, practitioners say that there is a stigma regarding treatment that influences adherence even before side effects appear. This stigmatization could be related to the informal and erroneous information that the patient has even before the start of treatment. Adherence is influenced by some kinds of contextual factors [15]. Thus, health professionals need to take these into account when trying to solve the problem. In previous studies in patients with generalized anxiety disorders, it has been reported that it is important that patients have adequate information to improve medication adherence [21]. As such, they should adequately explore the patient’s beliefs and provide the appropriate information. In this regard, the SDM process, which is associated with patient-centered care (PCC), is a useful and necessary method to achieve success [30]. Professionals have identified this tool as being useful and desirable in daily practice, but they have not identified it as a facilitator. They consider that, due to the pressure on the system and the lack of time, it is not always possible to put it into practice. However, the literature suggests that when patient involvement in decision-making leads to progress in the patient’s condition, the professionals are more efficient in terms of time management [30]. By contrast, it is noteworthy that some professionals have a misconception of what considering the patient’s preferences means. If it has been possible to adequately explore the patient’s preferences and, after that, the patients say that they prefer to leave the therapeutic decision in the hands of the health professional, this is also considered PCC. Furthermore, patients have identified SDM as a facilitator, although they recognize that they have not always been informed about the treatment by their doctors. This inconsistency between the professionals’ perception and the obtained results, as well as the misconception of what PCC means, shows the need to train professionals about the benefits of using this approach.

Although one of the best-known antidepressant-induced side effects is sexual dysfunction [31,32,33], this was not commented on by the patients and only one of the professionals pointed out its importance. Considering the importance of the sexual sphere, it could be considered that there was not enough confidence nor the appropriate context, due to the group dynamics, to address it adequately. However, it is important to remember that patients with depressive disorders, by definition, present low mood, are immersed in hopelessness, and their sexual behavior is often affected by the disease itself, so this may be a side effect, which is not easily identifiable by patients.

Regarding the origin of the patients’ beliefs about drugs, the professionals said that patients have access to quality information. They do not identify access as a problem since it is available on the internet. However, a digital divide remains despite the more generalized use of the internet, which is still not universal. It has been reported that the differences between the skills and the type of use are widening, even in the most advanced countries [32], and this is also true for health literacy [34,35]. This consideration, which professionals have about access to quality information, could affect the way in which the patient’s care plan is approached and how the patient adheres to it. Thus, the possibility of improving this preconception that professionals have is recommended in order to improve their awareness in this regard as well as emphasizing the importance of community health literacy.

As regards the patients’ perception, there was the following paradox: the clinical improvement with drugs was identified as both a facilitator and a barrier. Sometimes, patients admitted that they discontinued treatment on their own because they felt better. They considered that it was no longer needed. On other occasions, patients reported that they became more adherent when they improved, because they associated it with taking the medication. This dichotomy is found in other studies and reflects the strong influence of the individual patient’s treatment experience [25,36].

Another factor identified as a barrier by both groups was the involuntary lack of adherence due to forgetfulness. This phenomenon is especially true in psychiatric disorders [37]. Thus, the increasingly necessary use of reminders to reduce non-adherence is clear. A review has recently been conducted in which free and quality mobile phone applications have been shown to reduce non-adherence because of forgetfulness [38]. These mobile phone apps could be used in patients with depression. In addition, patients have identified the duration and severity of the disease as barriers, and this perception coincides with the results of quantitative studies [16,39]. There are other barriers identified in other studies, such as cost or access to the health system [40,41], which did not appear in the present study probably because Spain has a public health care system.

Social support is a facilitator of adherence identified by both groups. Specifically, family support was considered key by patients. In previous studies, it has been concluded that having social support leads to the improvement of medication adherence in chronic diseases [42], as well as to the improvement of depressive symptoms [43]. In addition, the doctor–patient relationship has been mentioned as another facilitating factor, this factor has been identified in the literature [44]. The trust established between the professional and the patient has been postulated as a fundamental pillar of therapeutic adherence.

By contrast, the professionals point out that poor management of depressive disorders in PC means a greater work overload for them. They highlight the lack of time, in the case of psychiatrists, and the lack of training in psychotherapy as one of their main limitations. However, PC professionals have even less time per patient and do not necessarily have more training in psychotherapy than psychiatrists. This perception can be explained as being the possible consequence of the fragmentation of the Spanish health system into medical specialties that hinders adequate coordination between different care levels and integration of patient care in practical terms. However, this issue was not mentioned by the patients, and the point of view of PC professionals is explored in the present study. Thus, the authors believe that future studies are necessary to include these actors in the equation.

The present study found that both professionals and patients mostly agree in identifying the factors that influence adherence. The results show the presence of many coincidences in the understanding of the problem between both groups. The findings of this exploratory research with the collected qualitative data have been taken into consideration for the design and application of a multicomponent strategy, addressed to both patients and psychiatrists, and aimed at improving adherence to pharmacological treatment in patients with depression, whose effectiveness and cost-effectiveness is currently being assessed by an RCT [22]. The fact of having considered the point of view of both professionals and patients leads the authors to adopt a positive position regarding the success of this strategy.

### 4.1. Consistency of Main Results with Previous Studies

The authors have found consistency between their results and those obtained in similar studies. Chon et al. [45] identified the following possible approaches to manage non-adherence to antidepressants: building alliances with patients and building patient support networks. As we have seen, these possible strategies have also been identified in the present study. By contrast, van Geffen et al. [46], in a similar study with patients under antidepressant treatment, have identified the appearance of side effects as one of the main causes of treatment discontinuation, as well as the need for shared decision-making to improve adherence. These factors have also been identified in the present study.

### 4.2. Limitations

The main limitation of this study is the size of the sample. It was not possible to reach data saturation in all the topics addressed here, and the degree of saturation of the different topics has been variable. Related to this, the study sample was small due to the pandemic context in which it was developed. The social distancing sanitary measures that were imposed during this period only allowed the meeting of groups of people of less than ten members. Thus, the focus groups were limited to a maximum number of eight patients, since this is considered the minimum number necessary to form a focus group. The moderator and an observer were added to this number, and the group could have ten members, which was the maximum allowed by the COVID-19 restrictions in place in Spain at the time. Furthermore, only patients and professionals from the Canary Islands, Spain, were included. Although the authors know that there are certain worldwide constant factors, different geographical and cultural variables could also influence therapeutic adherence, and it was not possible to consider these due to the sample characteristics. 

Finally, the pharmaco-biographical trajectory of the patients could be related to therapeutic adherence, but this is unknown. This limitation requires a more individualized and more in-depth approach to provide this information. However, it was not possible to do this because of lack of time and resources.

## 5. Conclusions

This study suggests the need to promote a good doctor–patient relationship and adequate patient information about depression and its treatment to increase their engagement in the recovery process. Furthermore, an SDM model should be adopted. Therefore, practical training of clinicians in this approach and other possible information tools to help patients overcome adherence barriers would also be necessary. In addition, it is important to promote a social support network and to maintain contact with family members to improve therapeutic adherence in patients with depressive disorders. Organizational aspects, such as adequate coordination between healthcare levels and more time in clinical visits, would also contribute to achieving these objectives.

## Figures and Tables

**Table 1 ijerph-19-16788-t001:** General characteristics of participants.

Participants of the Focus Groups (Patients and Informal Caregivers)		N = 11
**Age, mean (SD)**		50.72 (9.39)
**Age range**		38–65
**Gender, n (%)**		
	Women	8 (72.73)
	Men	3 (27.27)
**Informal caregivers**		2
**Patients of the focus groups**		**N = 9**
**Age, mean (SD)**		52.11 (9.81)
**Age range**		38–65
**Marital status, n (%)**		
	Married/with a partner	5 (55.55)
	Single/separated/widowed	4 (44.44)
**Gender, n (%)**		
	Women	6 (66.67)
	Men	3 (33.33)
**Educational level, n**		
	No formal education/incomplete primary education	3
	Primary education	1
	Lower secondary education	1
	Higher secondary education	4
	University education	0
**Diagnosis, n (%)**		
	F32-F33 Major depressive disorder	8 (88.89)
	F34 Dysthymic disorder	1 (11.11)
**Diagnosis duration, mean (SD)**		5.47 (6.65)
**Total antidepressant drugs, mean (SD)**		2
**Antidepressant drug range**		1–4
**Adherence to antidepressant treatment (Sidorkiewicz), mean (SD)**		6.78 (2.77)
**Adherence to antidepressant treatment (Sidorkiewicz), n (%)**		
	Yes	1 (11.11)
	No	8 (88.89)
**BDI-II, mean (SD)**		37.2 (9.55)
**Participants in the semi-structured interviews (psychiatrists)**		**N = 6**
**Age, mean (SD)**		39.43 (11.83)
**Age range**		29–63
	Women	3 (50)
	Men	3 (50)
**Island**		
	Gran Canaria	4
	Tenerife	2

BDI-II: Beck Depression Inventory-Second Edition; n: number of patients; NA: not applicable; SD: standard deviation; Sidorkiewicz: Sidorkiewicz adherence instrument.

**Table 2 ijerph-19-16788-t002:** Individual characteristics of participants of focus group (patients and caregivers).

Participants of the Focus Groups	Age	Gender	Diagnosis	Antidepressant Drugs (n)	Years Since Diagnosis
Patient 1	50	M	Dysthymia	4	2
Patient 2	42	W	Major depressive disorder	1	3
Patient 3	52	W	Major depressive disorder	1	6
Patient 4	38	W	Major depressive disorder	2	2
Patient 5	48	M	Major depressive disorder	1	2
Patient 6	47	W	Major depressive disorder	2	1
Patient 7	64	M	Major depressive disorder	1	22
Patient 8	63	W	Major depressive disorder	2	6
Patient 9	65	W	Major depressive disorder	3	1
Caregiver 1	42	W	Informal caregiver of patient 1	NA	NA
Caregiver 2	47	W	Informal caregiver of patient 5	NA	NA

M: man; n: number; W: woman; NA: not applicable.

**Table 3 ijerph-19-16788-t003:** Individual characteristics of the participants in the semi-structured interviews (psychiatrists).

Participants in the Semi-Structured Interviews	Age	Gender	Professional Features
Psychiatrist 1	48	M	Specialist doctor
Psychiatrist 2	52	M	Specialist doctor
Psychiatrist 3	37	W	Specialist doctor
Psychiatrist 4	47	W	Specialist doctor
Psychiatrist 5	63	W	Specialist doctor
Resident doctor 1	29	M	Doctor in the fourth year of residency training to become a psychiatrist

M: man; W: woman.

**Table 4 ijerph-19-16788-t004:** Modifying factors of therapeutic adherence identified by professionals and patients.

Dimensions	Factors	Perspective and Illustrative Quotes	
		Barriers	Facilitators
**Health system**	**Coordination with PC**	**Psychiatrists** *“The biggest barrier may be in primary care. It seems that primary care physicians are often better at first addressing hypertension training or diabetes problems or anything other than treating depression”; “There are depressions that are very easy to treat and that is also another reason for overloading psychiatrists with work. (…) There are a lot of patients who are not even minimally treated by their family doctors before referral to a psychiatrist.” *(Psychiatrist 2)** *“I think the main problem is when the diagnosis is not correct. In other words, if you are scrupulous about the psychopathology of depression, you will apply a treatment that will be effective, but many times there are patients who are diagnosed with depression when they are not depressed. (…) Generally, the first diagnosis is made by a primary care physician (…) the diagnosis is not always the right one”; “Primary care doctors have five or seven minutes per patient. Many times, they prescribe a standard medication.* (Psychiatrist 3) *“I think that a PC physician prescribing an antidepressant treatment for 6–12 months before referring them to a psychiatrist is too much.”* (Psychiatrist 4)	**Psychiatrists** *“However, when the patient is treated by a psychiatrist, then we have more time to get to know the patient and the treatment is more tailored.”; “Interventions to promote adherence to treatment, either with group activities or with follow-up by phone (could improve adherence).”; “It (motivational interview) is useful, but (…) you have to be trained. It is true that I have been half trained in motivational interviews. The truth is that I need the second half*.” (Psychiatrist 3)
	**Accessibility and availability of professionals**	**Psychiatrists** *“There is a lack of time for professionals to explain, not only what is happening, but what the therapeutic possibilities are”; “The system is not oriented so that the patient can receive regulated psychotherapy (because of a lack of psychologists)”; “If there is not enough time to attend to a patient, there is no motivational interview or anything, because you cannot dedicate yourself to them calmly”* (Psychiatrist 3) *“Psychotherapeutic work is harder, that is, with the person, trying to understand their situation… These activities are more typical of psychology, but the conditions are not met, especially in the hospital setting”;* *“Psychologists, due to their scarcity, cannot apply the therapies as they should and that ends up with overloading*.” (Psychiatrist 2) *“In the end, the psychiatrist ends up trying to do psychotherapy, without being the right professional, because the right professional should be a psychologist to do therapy properly*.” (Resident doctor 1) *“If you are a psychiatrist, (psychotherapy) needs complementary training, and not all psychiatrists have it. It depends on the interest”; “I can’t do it with everyone (psychotherapy), I don’t have time.”* (Psychiatrist 5)	**Patients** *“I think it would be necessary to provide hospitals with more staff. I think we would reduce the treatment a lot if the doctors were not so overworked and could, perhaps, treat the illness in another way.”* (Patient 8)
			**Psychiatrists** *“I think it is essential that patients are able to consult doubts to achieve adherence, that would improve adherence.”* (Psychiatrist 4) *“A greater availability of consultation by psychiatry and the availability of more frequent check-ups (would improve adherence).”* (Psychiatrist 2)
	**Information available to the patient**	**Psychiatrists** *“I admit that sometimes it’s easier for me if they come and don’t object… I am a bit ambivalent about that.”* (Resident doctor 1). *“People search on the internet. Another thing is that they search on the correct website”; “It’s a bad sign when a patient comes with what the neighbor told him/her. That’s a bad sign, it’s worse than googling”;* “*The information is accessible (…). On professional websites, Spanish Society of Psychiatry and Ministry of Health, there are specific guides on depression. In other words, access is not the problem”; “New patients (patients who have not received information) who come for the first time to receive a treatment generally do not cause problems, I explain the treatment and that’s it.”* (Psychiatrist 5). *“I don’t prefer the savvy patient, who knows everything, who has read… I like to explain a little bit to them, but today, when you mention serotonin, they say yes, yes, serotonin sounds familiar to me.”* (Psychiatrist 2)	**Patients** *“I received all the information. He told me what each thing was for, how I was going to take it, how I was going to start, and he explained everything to me… He explained everything to me, the entire medication process.”* (Patient 8)
			**Psychiatrists** *“I prefer they come informed, and they consult with me about their doubts, because they can sometimes come misinformed…”* (Psychiatrist 1). *“I think that information prior to treatment is essential (to promote adherence). If you explain everything to them before giving them a treatment, and they understand what the process is going to be, they accept it many times”; “My experience is that it depends on the information. Many times, it is not entirely correct or the websites where patients find information are not entirely reliable. Then, they come with a bad expectation.”* (Resident doctor 1) “*I prefer informed patients, well-informed patients, because I think that teamwork is necessary. At least, we are two experts*.” (Psychiatrist 5)
	**Doctor-patient relationship**		**Patients** *“In that aspect (adherence), I am super motivated by the doctor. I totally trust her”; “I think that medical support is essential. It is (depression) something that we don’t understand, we can’t find an explanation for it, and so you have to seek help from people who are trained in the matter, who know, to start getting out of it. Then, of course, you also have to do your bit, that is, you cannot wait for the medication to take effect, but you also have to put in a little on your part.”* (Patient 9) *“I have not read the prospects; I fully trust my psychiatrist*.” (Patient 1) *“The help of a psychiatrist has been very good for me. The advice he gives you, the guidelines you can follow.”* (Patient 10)
			**Psychiatrists** *“The therapeutic link, that is, the trust you have with the doctor who treats you (the psychiatrist or the primary care doctor) (is an important adherence factor).”* (Psychiatrist 3) *“If the patient normally trusts you, if you give him the option to consult doubts and such, he usually pays attention to you, but it is important that he trusts you.”* (Psychiatrist 5)
	**SDM**	**Psychiatrists** *“There are many patients who are not very interested in that (SDM), but what they simply want is for the doctor to prescribe them something and they don’t even get into discussing anything.”* (Psychiatrist 1) *“I try (do SDM), but I recognize that I cannot apply regulated psychotherapy (due to lack of time)”; “Well, it is true that I do not always do it perfectly. Sometimes the time is pressing and well*…” (Psychiatrist 4) *“Few psychiatrists say, ‘we would have this treatment and that and you choose’”; “Many times, they leave the decision to us (professionals), even if you tell them, they tell you, ‘Whatever you see best, I don’t understand.’”* (Resident doctor 1) *“This has to be done, I can’t tell you if it is done regularly, but it has to be done”; “There is a lack of time for professionals to explain not only what is happening but also what the therapeutic possibilities are”; “It is true that there are colleagues who feel more comfortable with paternalism:—‘you have to take this because I say so’”; “The time, the lack of time. If you have little time, you cannot apply shared decision-making because it requires time*.” (Psychiatrist 3)	**Patients** *“Yes, the psychiatrist and I were talking about it, and she listened to me. Then, we made the decision together”; “In my case, she explained the pros and cons to me, but since it’s an issue I don’t understand, I take the medication, and then if I feel bad, I ask her to change it*.” (Patient 3)
			**Psychiatrists** *“If you give the patient options that are more appropriate to their preferences, I think that it becomes the main positive conditioning factor for adherence”; “I really like to involve the patient in treatment options.”* (Psychiatrist 2). *“This has to be done (SDM), I can’t tell you if it is done regularly, but it has to be done”; “There are patients who ask you this (pseudoscience). I think that the patient’s decision must be respected as long as he is well informed”; “I would take into account the functionality of the patient, and by functionality, I mean work or daily living activities (…), and then I would also take into account the patient’s preferences.”* (Psychiatrist 3) *“Involving the patient is essential in the improvement process.”* (Psychiatrist 4) *“Especially in adherence (it has a positive influence), when a patient feels that they have decided on their treatment, that they are not obliged, they take it more easily.”* (Psychiatrist 5). *“I think the decision is made by the patient, it is clear, no matter how many pills you prescribe.”* (Resident doctor 1)
**Patient, family, and socioeconomic environmental**	**Socioeconomic and family context**	**Patients** *“Sometimes, when you get a very severe depression and you don’t recognize yourself, nor do your relatives recognize you, because you seem (to be) another person.”* (Patient 8)	**Psychiatrists** *“When there is family support, there is always greater adherence, because the family is there, saying ‘come on, you have to take it, you have to put up with it.’*” (Resident doctor 1) *“I think there should be more ‘society’, more neighborhood associations, more sense of having a family, a society, which you and others belong to”* (Psychiatrist 4) *“Illness awareness (favors adherence). That they know that they are sick and that they want to get better seems fundamental to me.”* (Psychiatrist 4) *“Apart from the information, the involvement of family members (is important). Asking a family member to prepare the medication and supervise it. I have also used that strategy”; “The involvement of family members, telling a family member to prepare the medication and supervise it. Yes, I have also used that strategy”; “The information and involvement of relatives (is very important) for taking treatment”.* (Psychiatrist 3) *“Another influence is that someone is supervising, especially older people, (…), because it is also a determining factor for the patient to take the medication”.* (Psychiatrist 1)
**Disease and treatment itself**	**Depression**	**Psychiatrists** *“Severe depression depletes resources, but when it is a mild or moderate depression, it (it) requires a normal effort”; “Real depressions are those that require less work. On the other hand, depressions with personality disorders, with family and social problems are the ones that require the most work.”* (Psychiatrist 4) *“An endogenous depression hits you and you want to cure yourself, like any other disease. People who have an underlying neurosis do not always want to be cured, and the most difficult thing is being able to cure them when they don’t want to be cured; it’s quite complicated.”* (Psychiatrist 5) *“If we are treating a very, very severe case of major depression, (…) we have to prioritize the drug treatment before using another type of strategy.” (Psychiatrist 1)* *“Those depressions with social, anxiety, family factors (…) are the ones that involve more work. In my opinion, due to the lack of psychological support.”* (Psychiatrist 2)	**Patients** *“I have had depression before and I got better on my own, but in this case, I needed medication. It was impossible to get better alone; therefore, medication was essential in this case.”* (Patient 1) *“I don’t care about the side effects. I was lying on a bed; I did not care what the pills did to me. The fear is what will happen when I stop taking the pills.”* (Patient 3)
			**Psychiatrists** *“You have to be very meticulous, identifying what the characteristics of the patient are in all senses, that is, in the personal sphere, in the field of health, what are the other pathologies that they have”; “I would take into account the functionality of the patient, and by functionality I mean work or daily living activities (…), and then I would also take into account the patient’s preferences.”* (Psychiatrist 3)
	**Medication**	**Patients** *“The times to take it, there are many a day. It makes it difficult for me on a day-to-day basis, at work…”* (Patient 4) *“Those of us who take pills have the hope of one day being the same as before.”* (Patient 2) *“The delay of the therapeutic effect, which many times both in the sense of beginning the treatment and at the end. That there is that delay between when you start taking the treatment and its effectiveness once you have a relapse. The truth is that this favors non-adherence a lot…”* (Patient 9) *“Antidepressants do not have an immediate effect; therefore, they take time to take effect and you start taking that and say: ‘Oh my God, this doesn’t do anything for me, what am I taking?’”* (Patient 8) *“In my case, I did a lot of research on medication side effects and that made me ask the doctor to reduce my medication. This matters a lot to me.”* (Patient 3) *“It is true that the medication has helped me, but it is also true that it has some very bad side effects and that sometimes I can’t stand it, I can’t stand it.”* (Patient 10)	**Psychiatrists** *“Pharmacological treatment has some protocols, in other words, this part is not very complex.”* (Psychiatrist 2) *“The problem is that many of the patients are resistant (to treatment). (…) So, first, you have to investigate more, especially the psychopharmacological approach that has been used.”* (Psychiatrist 1)
		**Psychiatrists** *“The main reason for lack of adherence is side effects.”* (Psychiatrist 3)	

PC: primary care; SDM: shared decision-making.

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
