# Peer review of "Barriers and Facilitating Factors of Adherence to Antidepressant Treatments: An Exploratory Qualitative Study with Patients and Psychiatrists"

_ijerph, 2022, doi:10.3390/ijerph192416788_

Round 1
Reviewer 1 Report
I thought there were a couple of places you could make it a bit stronger. One was that you're a bit cryptic about the differences between the clinician and the anthropologist. You assure us twice that no "bias" slipped in, but didn't tell us much about that worry or why we should be confident that it didn't effect the conclusions. Maybe one or two sentences more? I'd also assume that all of this was translated from Spanish, but you don't mention much about that or if it presented any challenges for the interpretation.
I do think it would be possible to make somewhat stronger conclusions from the material, especially in terms of what the clinicians should do to better support adherence (or to understand if adherence is still the right path, if there's disagreement).
Reviewer 2 Report
de León et al examined here qualitative factors that influence the antidepressant medication adherence of Spanish patients, with the intervention of caregivers.
The MS is well-written, with minor spelling errors, and some truncated sentences (please provide full English revision when responding). de León et al bring a more-qualitative discussion about the factors tha influence the antidepressant adherence of Spanish psychological patients, which is very relevant, specially because it brings personal testimonies from them. However, from my point of view, MAJOR modifications/corrections are still necessary before acceptance:
(1) The authors have to reinforce the relevance of their study since it was based on a very limited sample size (maximum N = 11, in one interview session). I fully agree that the personal testimonies reinforce the impact of such a qualitative-driven work, but I suppose that comparisons with other similar (but more robust) studies would do a better job. Maybe a full new session/subtitle will be necessary for that.
(2) Authors should strength and discuss the impact antidepressant medication causes on sexual behavior and social interactions between peers. These are primal and instinctive behaviors that are directly influenced by these drugs. The limited sample size raised me the question if it is not some kind of social distance or simply isolation (in Canary Islands, pandemics side-effects, etc) that triggering such worrying testimonial.
REF: Lorenz T, Rullo J, Faubion S. Antidepressant-Induced Female Sexual Dysfunction. Mayo Clin Proc. 2016 Sep;91(9):1280-6. doi: 10.1016/j.mayocp.2016.04.033. PMID: 27594188; PMCID: PMC6711470.
Montgomery SA, Baldwin DS, Riley A. Antidepressant medications: a review of the evidence for drug-induced sexual dysfunction. J Affect Disord. 2002 May;69(1-3):119-40. doi: 10.1016/s0165-0327(01)00313-5. PMID: 12103459.
Taheri Zadeh Z, Rahmani S, Alidadi F, Joushi S, Esmaeilpour K. Depresssion, anxiety and other cognitive consequences of social isolation: Drug and non-drug treatments. Int J Clin Pract. 2021 Dec;75(12):e14949. doi: 10.1111/ijcp.14949. Epub 2021 Oct 14. PMID: 34614276.
Round 2
Reviewer 2 Report
I am glad the authors sufficiently improved the quality and scientific soundness in this updated version of their work. English corrections significantly increased the fluency of the text.
I consider the MS now ready for publication in IJERPH/MDPI
Thank you for the opportunity to review this work.